# Traumatic Anterior Cervical Disc Herniation Presenting as Severe Dysphagia

**DOI:** 10.3390/diagnostics13243644

**Published:** 2023-12-12

**Authors:** Jonghun Seo, Jeonghyun Oh, Pius Kim, Chang Il Ju, Seok Won Kim

**Affiliations:** 1Department of Neurosurgery, College of Medicine, Chosun University, Gwangju 61452, Republic of Korea; seo3622@naver.com (J.S.); gamechanger@chosun.ac.kr (P.K.); jchangil@chosun.ac.kr (C.I.J.); 2Department of Otorhinolaryngology-Head and Neck Surgery, College of Medicine, Chosun University, Gwangju 61452, Republic of Korea; jayoh@chosun.ac.kr

**Keywords:** dysphagia, esophagus, trauma

## Abstract

Due to the anatomical characteristics of the cervical spine, few cases of traumatic anterior cervical disc herniation have been reported in the literature. Here, we present a rare case of a traumatic anterior cervical disc herniation presenting as severe dysphagia. A 75-year-old male patient presented with severe dysphagia following an accident three days prior when he fell from a height of stairs. Cervical magnetic resonance (MR) imaging revealed a 1.3 × 1.0 cm extruded disc in the anterior aspect of the C4 level with the base at the C3–4 disc, which displaced the esophagus anteriorly. Esophagography revealed an extrinsic esophageal lesion that was considered to be responsible for the obstruction of the airway at the same level. He underwent a ruptured disc removal via the anterior approach. Preoperative dysphagia was resolved gradually after surgery, and he remained asymptomatic six months after surgery.

Traumatic cervical disc ruptures without other severe injuries are rare. Traumatic anterior cervical disc rupture is much rarer because the annulus fibrosis is thin in the posterior and posterolateral portions [1].

However, known cases of severe dysphagia resulting from hypertrophic anterior osteophytes have been increasing, and they are mostly seen in patients with diffuse idiopathic skeletal hyperostosis [2].

Due to the soft peristaltic movement of the esophagus, symptomatic dysphagia caused by anterior cervical disc herniation without hypertrophic osteophytes is not common. Moreover, symptomatic anterior cervical disc herniation after low-velocity accidents is extremely rare.

Here, we report a rare case of a traumatic anterior disc rupture presenting as severe dysphagia, which was successfully treated with surgical removal of the ruptured disc. A 75-year-old male was admitted to the emergency room (ER) with primary complaints of neck discomfort and severe dysphagia. 

He had a minor accident in which he fell from a height of stairs three days prior, and he was transferred from the local otolaryngology clinic for a detailed examination regarding his difficulty in swallowing. His complaints were initially limited to neck discomfort, which later on progressed to difficulty in swallowing liquids rather than solid foods (Figure 1). The neurological examination results were unremarkable. Simple radiographs and a computed tomography scan of the cervical spine revealed a loss of lordotic curvature but no segmental instability, osteophytes, or fractures. Magnetic resonance (MR) imaging of the cervical spine revealed a 1.3 × 1.0 cm sized extruded disc in the anterior aspect of the C3–4 disc space, which displaced the esophagus anteriorly (Figure 2). Esophagography revealed an extrinsic esophageal lesion that was considered to be responsible for severe dysphagia at the same level (Figure 3).

Because the patient did not respond to conservative treatment for severe dysphagia, he underwent ruptured disc removal via an anterior approach. In the surgical field, a prevertebral hematoma and soft disc material, which had ruptured through the ventral annulus at the C3–4 disc level, were identified (Figure 4). The surgical removal of the ruptured disc was performed without any complications, and there were no fibrotic adhesions. Cage insertion with instrumentation was not performed because no significant fractures, instability, or cord compression were detected during surgery. 

A final pathological examination confirmed the diagnosis. He was mobilized on the second postoperative day, and his severe dysphagia was resolved completely. Follow-up esophagography performed on the seventh postoperative day revealed a normal peristaltic esophagus (Figure 5). Because the annulus fibrosis is the thinnest posteriorly, the majority of disc herniations affect the posterior or posterolateral side of the disc structure. The anterior portion of the cervical disc has no neural structures and due to the soft peristaltic movement, the esophagus does not present with dysphagia until severe compression is applied. Therefore, anterior cervical disc herniation is almost always asymptomatic, and so there are few cases of severe dysphagia in the literature [3,4].

The exact mechanism through which anterior cervical disc herniation takes place is still unknown, as in our case. Kim et al. have reported that even a small osteophyte with disc herniation at the C5–6 level can cause dysphagia because of compression against the hard cricoid cartilage located anterior to the esophagus [5]. In our case, it is thought that the amount of ruptured disc material was relatively moderately sized; thus, when the ruptured disc compressed the esophagus against the hyoid bone due to the anatomical characteristics, this caused severe dysphagia. In summary, although anterior cervical disc herniation after low-velocity accidents is a very rare disease entity, it can potentially occur. Hence, anterior herniation of a cervical disc should be included in the differential diagnosis of severe dysphagia. Based on our experience, in cases where ruptured disc material causes severe dysphagia, the ruptured disc material should be removed.

## Figures and Tables

**Figure 1 diagnostics-13-03644-f001:**
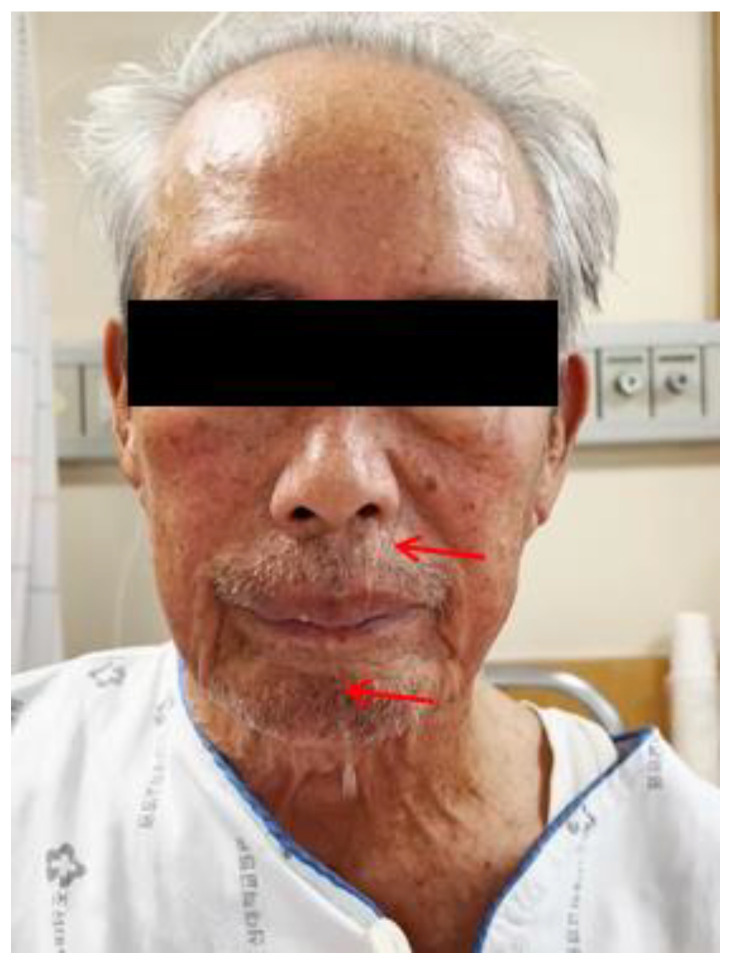
Water was seen to come out of his nose when he swallowed (arrows).

**Figure 2 diagnostics-13-03644-f002:**
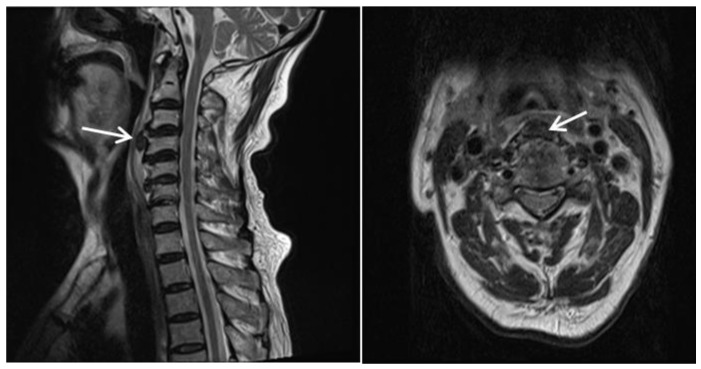
T2-weighted sagittal and axial magnetic resonance images show an anterior disc herniation and soft tissue swelling at the C3–4 level (arrows).

**Figure 3 diagnostics-13-03644-f003:**
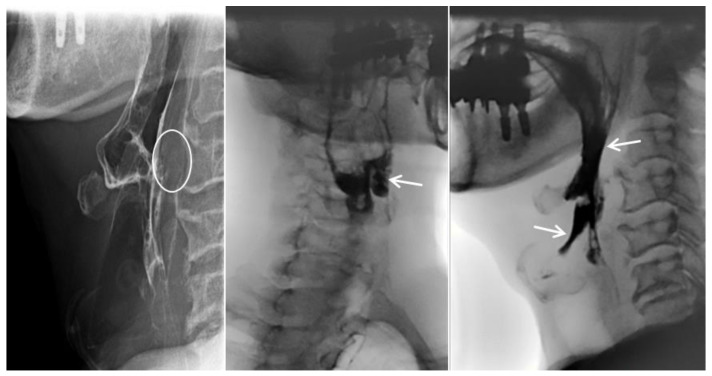
The esophagram revealed soft tissue swelling at the C3–4 level (circle), contrast retention in the laryngeal vestibule with aspiration, and reflux into the larynx (arrows).

**Figure 4 diagnostics-13-03644-f004:**
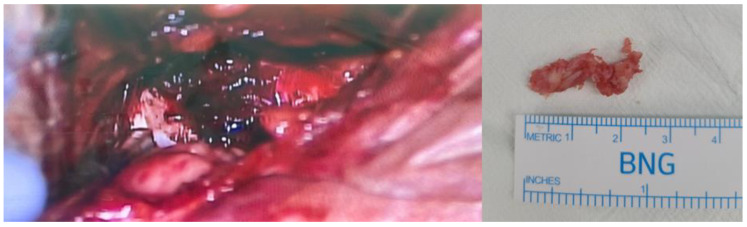
The surgical field is seen to reveal a prevertebral hematoma and a ruptured disc fragment.

**Figure 5 diagnostics-13-03644-f005:**
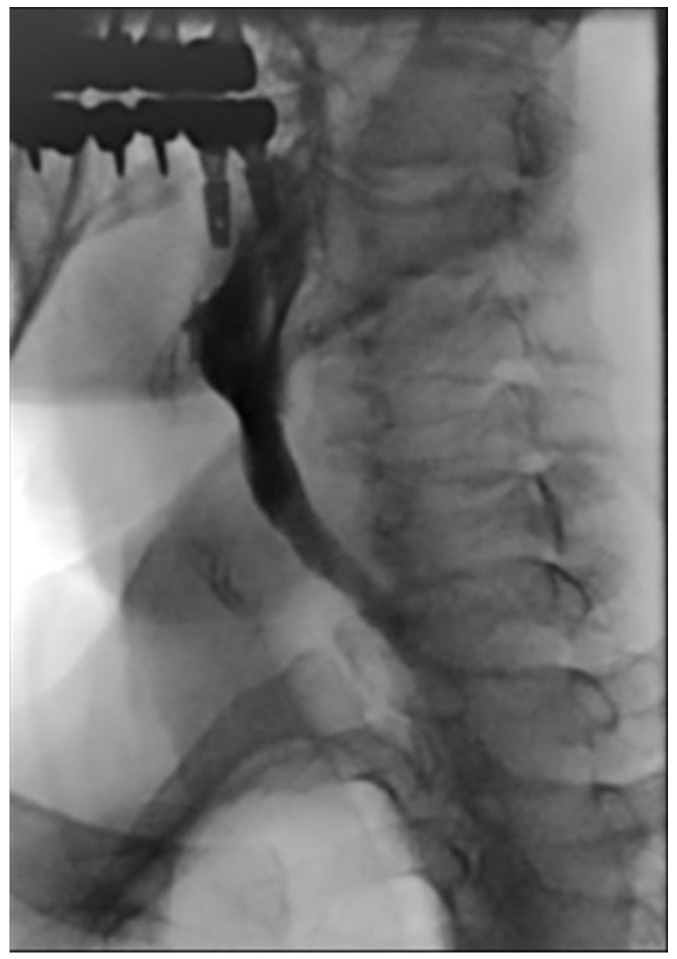
Esophagram taken seven days after surgery showed improvement in penetration and aspiration into the laryngeal vestibule; reflux into the pharynx was also shown to be improved.

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
