# Peer review of "Traumatic Anterior Cervical Disc Herniation Presenting as Severe Dysphagia"

_diagnostics, 2023, doi:10.3390/diagnostics13243644_

Round 1
Reviewer 1 Report
Comments and Suggestions for Authors
The case report is interesting.
The information reported in the text relating to surgical treatment is very limited. The surgical treatment should be better described.
Were there no indications for the placement of a cage?
Author Response
Cage insertion with instrumentation was not performed because no significant fractures, instability or cord compression were not detected during surgery. We have added this in the manuscript. Thank you very much for your comments.
Reviewer 2 Report
Comments and Suggestions for Authors
This case report provides a valuable contribution to the medical literature by detailing a rare instance of traumatic anterior cervical disc rupture resulting in severe dysphagia. It offers a comprehensive account of the patient's presentation, diagnostic procedures, surgical intervention, and postoperative outcomes.
The report is well-structured, systematically presenting the case from the initial symptoms and diagnostic findings to the surgical procedure and the patient's recovery. It effectively highlights the rarity of symptomatic anterior cervical disc herniation and emphasizes the significance of considering this condition in cases of severe dysphagia, despite its infrequency.
The inclusion of imaging results, figures, and references to existing literature strengthens the report's credibility and provides a clear visual understanding of the anatomical considerations and the extent of the injury. The explanations regarding the mechanisms underlying dysphagia in this context and the recommendations drawn from the experience offer practical insights for medical practitioners.
Overall, this case report is well-documented, informative, and contributes substantially to the understanding and management of traumatic anterior cervical disc ruptures leading to severe dysphagia.
Comments on the Quality of English LanguageThis case report provides a valuable contribution to the medical literature by detailing a rare instance of traumatic anterior cervical disc rupture resulting in severe dysphagia. It offers a comprehensive account of the patient's presentation, diagnostic procedures, surgical intervention, and postoperative outcomes.
The report is well-structured, systematically presenting the case from the initial symptoms and diagnostic findings to the surgical procedure and the patient's recovery. It effectively highlights the rarity of symptomatic anterior cervical disc herniation and emphasizes the significance of considering this condition in cases of severe dysphagia, despite its infrequency.
The inclusion of imaging results, figures, and references to existing literature strengthens the report's credibility and provides a clear visual understanding of the anatomical considerations and the extent of the injury. The explanations regarding the mechanisms underlying dysphagia in this context and the recommendations drawn from the experience offer practical insights for medical practitioners.
Overall, this case report is well-documented, informative, and contributes substantially to the understanding and management of traumatic anterior cervical disc ruptures leading to severe dysphagia.
Author Response
Thank you very much for your positive comments.